# Correlation between Blood Type 0 and Risk of Chronic Subdural Hematoma Recurrence: A Single Center Retrospective Cohort Study

**DOI:** 10.3390/brainsci13040567

**Published:** 2023-03-28

**Authors:** Motaz Hamed, Tim Lampmann, Abdallah Salemdawod, Harun Asoglu, Naomi Houedjissin, Marcus Thudium, Lakghomi Asadeh, Frederic Carsten Schmeel, Fabiane Schuch, Hartmut Vatter, Mohammed Banat

**Affiliations:** 1Department of Neurosurgery, University Hospital Bonn, 53127 Bonn, Germany; motaz.hamed@ukbonn.de (M.H.); tim.lampmann@ukbonn.de (T.L.); abdallah.salemdawod@ukbonn.de (A.S.); harun.asoglu@ukbonn.de (H.A.); naomi.houedjissin@ukbonn.de (N.H.); hartmut.vatter@ukbonn.de (H.V.); 2Center for Advanced Imaging Research, Department of Diagnostic Radiology and Nuclear Medicine, Marlene and Stewart Greenebaum Comprehensive Cancer Center, University of Maryland, Baltimore, MD 20742, USA; 3Department of Anesthesiology and Intensive Care Medicine, University Hospital Bonn, 53127 Bonn, Germany; marcus.thudium@ukbonn.de; 4Department of Neuroradiology, University Hospital Bonn, 53127 Bonn, Germany; asadeh.lakghomi@ukbonn.de (L.A.); carsten.schmeel@ukbonn.de (F.C.S.); 5Department of Neurology, University Hospital Bonn, 53127 Bonn, Germany; fabiane.schuch@ukbonn.de

**Keywords:** chronic subdural hematoma, ABO blood type, minor head trauma

## Abstract

Chronic subdural hematoma (cSDH) is a common disease in the neurological and neurosurgical world. The recommended treatment for cSDH patients with moderate or severe neurological symptoms is surgical evacuation, but cSDH frequently recurs. The patient’s ABO blood type may influence the outcome. This study aims to evaluate the correlation between cSDH recurrence and blood type O. We performed a retrospective analysis of the data of patients with cSDH who were surgically treated. Recurrence was defined as the need for re-operation within the first 12 weeks after the initial surgery. We analyzed standard demographic data, duration and type of surgery, ABO blood types, and the re-operation rate. Univariate and multivariate analyses were conducted. A total of 229 patients were included. The recurrence of hematoma was identified in 20.5% of patients. Blood type O was found to be significantly associated with cSDH recurrence leading to re-operation within 12 weeks (*p* = 0.02, OR 1.9, 95% CI 1.1–3.5). Thrombocyte aggregation inhibition and oral anticoagulants were not predictors of cSDH recurrence. Patients with blood type O in our cohort were identified to be at higher risk of cSDH recurrence and may, therefore, be a more vulnerable patient group. This finding needs further evaluation in larger cohorts.

## 1. Introduction

Chronic subdural hematoma (cSDH) is a common disease in the neurological and neurosurgical world. The incidence of cSDH increases with age [1]. This incidence in the population can be expected to increase over the next decades [2]. In addition to age, risk factors for cSDH include anticoagulation therapy, cerebral atrophy, alcoholism, and minor head injuries [3,4]. The clinical signs of cSDH vary between headaches and vomiting as a sign of raised intracranial pressure to hemispheric symptoms such as limb weakness, speech impairment, and gait disturbances or falls [5]. Surgical evacuation is the recommended treatment for patients with cSDH with relevant neurological symptoms, preferably by burr hole drainage and opening the dura to release the hematoma [6].

It has been suggested that the ABO blood group system has an impact on and correlations with neurosurgical pathologies [7,8,9]. A recent retrospective study demonstrated a significantly higher risk for patients with blood type A suffering from cSDH recurrence after the surgical evacuation of the hematoma [7]. The ABO blood type is expressed in different cell types, including red blood cells, platelets, and endothelial cells, affecting primary hemostasis and, therefore, has an influence on the risk of thromboembolic and hemorrhagic events by interacting with the von Willebrand factor (vWF) glycoprotein [10]. Blood type O is associated with lower plasma vWF levels and with a higher risk of increased bleeding and bleeding severity [11,12].

Given these characteristics of blood type O, this study aimed to evaluate whether this blood type was associated with a higher risk of cSDH recurrence after their primary surgical procedure.

## 2. Materials and Methods

### 2.1. Study Design and Inclusion Criteria

Data from all patients with cSDH aged > 18 years who had undergone primary hematoma evacuation between 2015 and 2020 at the Department of Neurosurgery of the University Hospital Bonn were entered into a computerized database (SPSS, version 25, IBM Corp., Armonk, NY, USA). Follow-up checks were conducted up to 3 months after surgery. Patients’ clinical information, including age, sex, site, the type of hematoma, surgical procedure, midline shift pre-surgery, neurological and functional status, Glasgow Coma Scale (GCS), blood type, intraoperative blood loss, the use of drainage, perioperative antithrombotic therapy, cSDH recurrence rate, and in-house complications were recorded. Every patient received a preoperative and postoperative CT scan.

We excluded all patients who had multiple simultaneous intracranial hemorrhages. Additionally, patients with incomplete data were excluded. There were 11 patients in our cohort.

### 2.2. Surgical Procedure

Surgical treatment involved one or two burr holes and irrigation with normal saline; the reflux of clear subdural fluid indicated hematoma removal. For gravity drainage, we placed a subdural drain with a feeding tube catheter, which we removed the second or third day after surgery, and after a follow-up, a CT scan confirmed a decrease in the heamatoma. This CCT was used later to compare whether the follow-up control experienced recurrence or not.

We distinguished three subtypes of cSDH: Hygroma as hypodens on CT and intraoperative showing a liquor-typical color. The chronic heamatoma with the color of old oil intraoperative and iso- or hypodens on CT. And finally, the subacute form; here, intraoperative fresh hematoma parts showed up as hyperdens on the CT as well.

If patients had taken antiplatelet agents preoperatively, they were operated on by substituting with tranexamic acid-one-time intraoperative 1 g and later continuously intravenously up to 24 h with a total of 3 g. When patients took oral anticoagulation, we compensated actively with the coagulation—depending on the substance—before rather than intraoperatively. Antithrombotic therapy occurred directly on the first day after the operation, with a low dose of heparin used as prophylaxis in the same procedure in all patient groups. Full therapeutic anticoagulation was routinely performed after the first control follow-up CT without recurrence; until then, a low dose of heparin was given as prophylaxis.

### 2.3. Radiological Evaluation

Recurrence was defined as progressive cSDH across the width of the bone with incipient or increasing midline displacement. Two independent radiologists performed a postoperative follow-up imaging assessment; the first radiologist was usually a radiologist outside our institute, where patients had their follow-up examinations. The second radiologist from our institute conducted the imaging during the outpatient presentation to us to decide on further procedures. In the case of disagreement about whether it was recurrence or not, we took the decision of our neuroradiologist from our institute because we all had the imaging (pre- and after surgery) and a comparison between all imaging. This study was conducted in accordance with the 1964 Helsinki declaration and was approved by the Ethics Committee of the University Hospital Bonn (reference no. 069/21). Informed consent was not sought as a retrospective study design was used.

### 2.4. Statistical Analysis

Data analysis was performed using SPSS (version 25, IBM Corp., Armonk, NY, USA) and Prism computer software packages. Categorical variables were analyzed in contingency tables using Fisher’s exact test. The Mann–Whitney U test was chosen to compare continuous variables as the data were mostly not normally distributed, while non-parametric data were summarized by median values (first quartile–third quartile). Results with *p* < 0.05 were considered statistically significant. Univariate analysis was conducted using Fisher’s exact test (two-sided) and the independent *t*-test. *p* values < 0.05 were considered to be statistically significant. In addition, in order to determine an independent predictor of cSDH recurrence in patients with surgically treated cSDH, a backward stepwise method was used to construct a multivariate analysis using a binary logistic regression, and again with *p* < 0.05, which was considered statistically significant.

## 3. Results

### 3.1. Patient Characteristics and Demographic Data

A total of 229 patients with 284 cSDHs were treated surgically and included. Table 1 shows the baseline data. CSDH did not recur in 79.5% of our patient population, whereas it did recur in 20.5% of our patients.

### 3.2. Patient-Related and Disease-Related Factors Associated with Recurrence of cSDH

Figure 1 and Figure 2 show the distribution of ABO blood groups among our patient cohort with and without the recurrence of cSDH.

In our study, we found that cSDH recurred most frequently in patients with blood type O (*p* = 0.004). Additionally, in the univariate analysis, patients with alcohol consumption were associated with cSDH recurrence. We found no relevant differences between the subtypes of recurrent cSDH and blood type.

Only blood type O was significantly associated in the multivariate analysis with cSDH recurrence, leading to re-operation within 12 weeks (*p* = 0.02, OR 1.9, 95% CI 1.1–3.5). Platelet aggregation inhibition and oral anticoagulants were not significant predictors of cSDH recurrence.

## 4. Discussion

The recurrence of cSDH continues to be a challenge in clinical practice. Current research indicates a cSDH recurrence rate of 10–20% [7,13,14]. In our cohort, the rate of recurrence was 20.5%.

CSDH particularly affected older patients, who had an average age of 78–80 years in our study. In an aging society, the incidence of cSDH is expected to increase over the next few decades with the consequence of a growing healthcare burden and financial impact [2].

The ABO blood type system influences primary hemostasis and, therefore, the risk of thromboembolic and hemorrhagic events by interacting with the vWF glycoprotein [10]. Several chemokines have been determined in cSDH fluid after trepanation surgery, including vWF [15]. In a rat model of cSDH, the balance between levels of pro-inflammatory cytokines (IL-6, IL-8 and TNF-α) in the early stages and levels of anti-inflammatory cytokines (IL-10, IL-13) and in the subacute and chronic phases influenced the introduction and absorption of the hematoma; among these, the upregulation of vWF was associated with recovery from cSDH [16]. While blood type O is associated with lower plasma vWF levels [11], we do not know if, for example, it could be associated with a decrease in vWF levels in the subdural cavity and a higher risk of cSDH recurrence. Considering our hypothesis on whether there is an association between the recurrence of cSDH and blood group, our work could prove this association. We tried to find a reason for this by researching the literature; we could not find any work that could explain this association. We assume that perhaps there are factors at the cellular level that favor this association, such as receptors in blood type O, but this is much more suggestive than proven. We think our work provides a basis for such studies at the molecular and cellular levels.

On the one hand, an association between blood group O and increased bleeding and bleeding severity was also shown [12], suggesting a pro-hemorrhagic role as well as a significant ABO group-specific difference in nonvascular primary hemostasis [17]. On the other hand, different studies concerning cerebral and non-cerebral hemorrhages did not demonstrate a significant difference among patient groups depending on their ABO blood type [18,19]. However, since cSDHs do not follow major acute bleedings but involve a chronic inflammatory process, including angiogenesis and microhemorrhages, a comparison with studies of acute hemorrhagic events seems to be insufficient [16]. The research study group of Hamou et al. recently published a study on the risk factors influencing the recurrence of cSDH; here, the configuration and density of Hematoma were the main factors, and blood type was not considered [20]. A variety of works have been presented concerning other risk factors of cSDH recurrence, but little is known so far about the relationship between the ABO system and cSDH recurrence [21,22,23,24,25,26]. To the best of our knowledge, this work is one of few to address this issue.

A recent study in Japan suggested that patients with blood type A have a significantly higher risk of cSDH recurrence after the surgical evacuation of the hematoma [7], but no further investigations have been performed, nor were there any studies in a European country. As the distribution of the ABO blood groups varies in different global locations, studies with different patient populations are necessary to determine how applicable these results are in different circumstances and to distinguish possible deviations [7,8,9,27]. The fact that we identified German patients with blood type O to be at a significantly higher risk of cSDH recurrence contrasts with the results of other countries.

Interestingly, the study of Yunwei et al. showed a completely different result than we found in our cohort, since they found that the ABO system had no effect on the recurrence of cSDH [28].

Our results should be tested in a larger, prospective study, and further investigations on the underlying pathways of this mechanism are necessary.

### Limitations

There are several limitations to be recognized in this study. Data acquisition was affected retrospectively in a single center and without randomization. Although surgical evacuation by burr hole drainage is a standard procedure, it can be expected to vary between different neurosurgeons. Additionally, the number of patients is limited because patients with an unknown ABO blood type were excluded from the study. Our data represent a short-term follow-up and are not a long-term outcome.

## 5. Conclusions

Our study found that patients with blood type O were at a higher risk of cSDH recurrence. Since blood type O occurs frequently in the general population, it indicates that a major group may be more vulnerable to cSDH recurrence. This finding needs further evaluation in larger cohorts.

## Figures and Tables

**Figure 1 brainsci-13-00567-f001:**
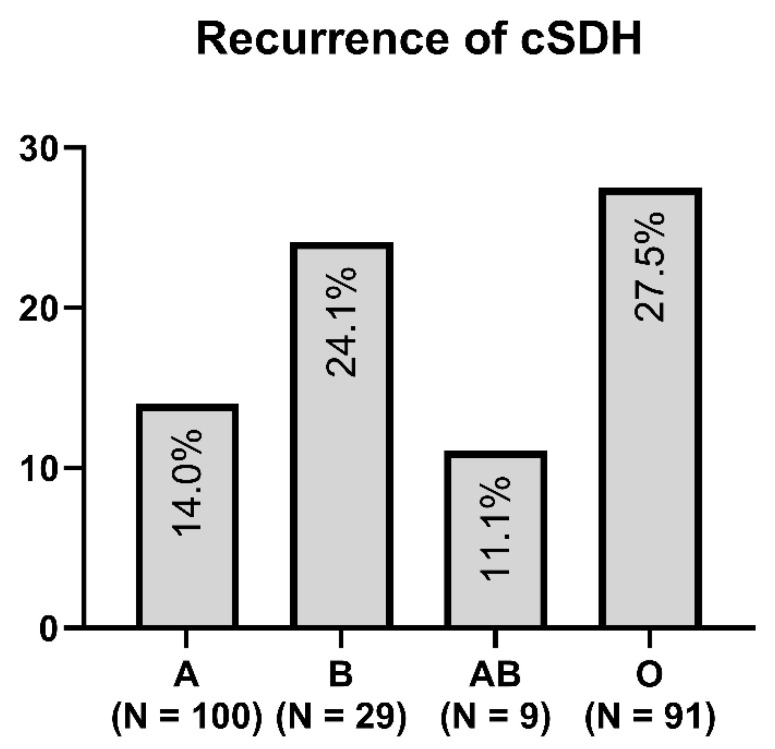
Distribution of blood type in the recurrence group of cSDH.

**Figure 2 brainsci-13-00567-f002:**
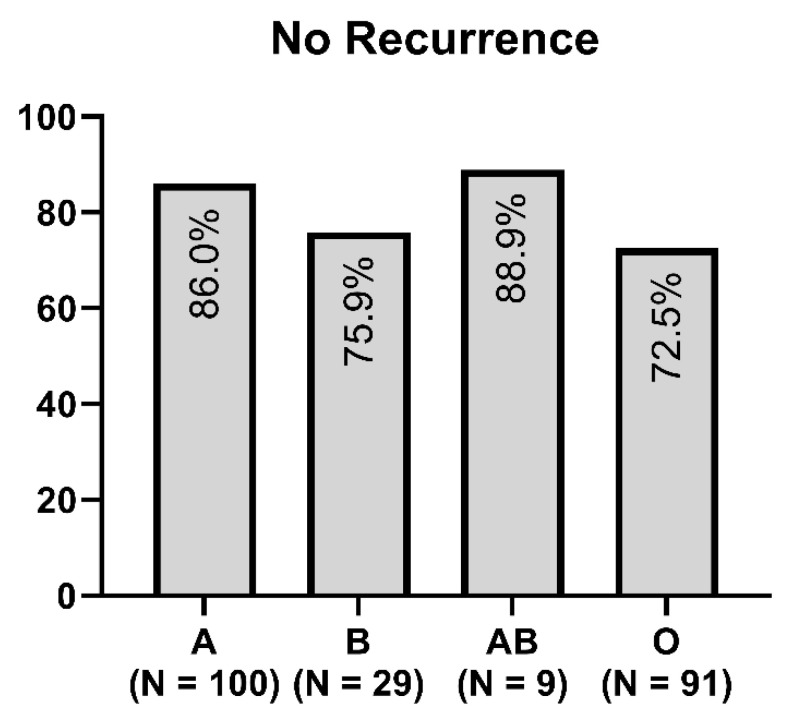
Distribution of blood type in the non-recurrence group of cSDH.

**Table 1 brainsci-13-00567-t001:** Baseline patient data and univariate analysis of characteristics and surgical procedures.

Total (N = 229)	cSDH without Recurrence 182 (79.5%)	cSDH with Recurrence 47 (20.5%)	*p* Value
Age (yrs.), median [q1–q3]	80 [78–83]	78 [76–81]	0.901
Sex			0.729
Female	60 (33.0%)	14 (29.8%)	
Male	122 (67.0%)	33 (70.2%)	
Site of hematoma			0.838
Left	69 (37.9%)	20 (42.6%)	
Right	61 (33.5%)	15 (31.9%)	
Both	52 (28.6%)	12 (25.5%)	
Type of hematoma			0.739
Chronic	147 (80.8%)	40 (85.1%)	
Hygroma	1 (0.5%)	0 (0%)	
Sub-acute	34 (18.7%)	7 (14.9%)	
Midline shift pre-surgery	137 (75.3%)	35 (74.5%)	1.000
Blood loss intraoperative, mL, median [q1–q3]	85 [82–89]	100 [94–101]	
Length of stay, in days, median [q1–q3]	6 [4–7]	5 [3–6]	
Postoperative subdural drain	170 (93.4%)	43 (91.5%)	0.748
Anti-thrombotic therapy			
ASS/Plavix	72 (39.6%)	19 (40.4%)	1.000
Oral anticoagulation	51 (28.0%)	12 (25.5%)	0.909
Neurological symptoms pre-surgery			
Epilepsy	20 (11.0%)	7 (14.9%)	0.0654
Headache	53 (29.1%)	13 (27.7%)	1.000
Vigilance	42 (23.1%)	12 (25.5%)	0.704
Paresis	87 (47.8%)	26 (55.3%)	0.414
In-hospital complications			
Pneumonia	5 (2.7%)	2 (4.3%)	0.714
UTI	6 (3.3%)	1 (2.1%)	0.890
Mortality	4 (2.2%)	2 (4.3%)	0.605
Blood type			
A	86 (47.2%)	14 (29.8%)	0.614
B	22 (12.1%)	7 (14.9%)	0.889
AB	8 (4.4%)	1 (2.1%)	0.726
O	66 (36.3%)	25 (53.2%)	0.004 *
Comorbidities			
Hypertonia	123 (67.6%)	30 (63.8%)	0.728
Malignancy	23 (12.6%)	11 (23.4%)	0.104
Diabetes mellitus	36 (19.8%)	15 (31.9%)	0.080
Cardiac	80 (43.9%)	20 (42.5%)	0.909
Alcohol	1 (0.5%)	4 (8.5%)	0.007 *

Categorical variables are shown as a number (%) and continuous variables as the median [interquartile range]. ASS: Acetylsalicylic acid. cSDH: chronic subdural hematoma. q1–q3: First quartile–third quartile. UTI: urinary tract infection yrs.: Years. * *p* < 0.05: statistically significant.

## Data Availability

All data generated or analyzed during this study are included in this published article.

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
