# Peer review of "Correlation between Blood Type 0 and Risk of Chronic Subdural Hematoma Recurrence: A Single Center Retrospective Cohort Study"

_brainsci, 2023, doi:10.3390/brainsci13040567_

Round 1

Reviewer 1 Report

The authors conducted a retrospective study to assess the underlying etiology of CSDH recurrence. They found that blood type O had the highest recurrence rate. The manuscript is well-written

I have the following comments/questions: 

Patients with CSDH are also elderly, and the average life expectancy in Germany is probably around 80 years. How do you determine the possibility that some patients dropped out due to death or other reasons before recurrence? Also, even if the patient did not die, do you consider the possibility that the patient went to another hospital at the time of recurrence, etc.?

How did you distinguish the recurrence of CSDH from bleeding as a postoperative complication? 

What is the difference between those studies (Hirai et al. and Ou et al.) and yours?

Please provide whether tranexamic acid or other hemostatic agents were used post-operatively. 

Author Response

The authors conducted a retrospective study to assess the underlying etiology of CSDH recurrence. They found that blood type O had the highest recurrence rate. The manuscript is well-written. 

                Author´s response: We thank the Reviewer for this comment.

I have the following comments/questions: 

  • Patients with CSDH are also elderly, and the average life expectancy in Germany is probably around 80 years. How do you determine the possibility that some patients dropped out due to death or other reasons before recurrence? Also, even if the patient did not die, do you consider the possibility that the patient went to another hospital at the time of recurrence, etc.?

Authors’ response:

The reviewers' criticism is valid and represents an important issue in the care of cSDH, especially after primary surgery. Since our patients cohort are around 80 years old, most of them will be transferred to acute care geriatrics after our surgical treatment, these hospitals are in our care sector and present all patients to us on our recommendation for follow-up control. A few percent go home and come from the neighboring region from our neurosurgical institute, so the connection to us postoperatively is very safe. We had of course excluded patients where a lot of data was missing, possibly gone to another practitioner or deceased. These patients were in our cohort of 11 patients. We have now added this important information to our exclusion criteria into methods section. Furthermore, we have added the limitation that our data represent a short follow-up and are not a long-term outcome.

Changes to the manuscript:

Methods section: Additionally, patients with incomplete data were excluded. These patients were in our cohort 11 patients.

Limitations section: Our data represent a short-term follow-up and are not a long-term outcome.

  • How did you distinguish the recurrence of CSDH from bleeding as a postoperative complication? 

Authors’ response:

Our house standard was that after each primary neurosurgical removal of the cSDH, a CT control after three days, or a control before discharge or transfer to the acute care geriatrics. This CCT is used later to compare whether the follow-up control is recurrence or not.

Changes to the manuscript:

Methods section: For gravity drainage, we placed a subdural drain with a feeding tube catheter, which we removed the second or third day after surgery and after a follow-up CT scan confirmed a decrease in the heamatoma. This CCT is used later to compare whether the follow-up control is recurrence or not.

  • What is the difference between those studies (Hirai et al. and Ou et al.) and yours?

Authors’ response:

The Hiari et al study group found in their Japanese cohort that recurrence was significantly associated with blood type A, but he also found that there was a trend with blood type O, but it was not relevant. In contrast, however, the Chinese study group Qu et al. found that cSDH was more likely to be associated with blood type B, and the group in their cohort was significantly older than the patients with blood type A and O. Our finding –german cohort- that the blood group O was associated with recurrent cSDH.

  • Please provide whether tranexamic acid or other hemostatic agents were used post-operatively

Authors’ response:

We expanded our surgical procedure under methods section

Changes to the manuscript:

If patients had taken antiplatelet agents preoperative, they were operated on by substituting with tranexamic acid -one-time intraoperative 1 gram and later continuously intravenously up to 24 hours with a total of 3 grams. When patients taked oral anticoagulation, we compensated actively the coagulation -depending on the substance- before and not intraoperative. Antithrombotic therapy was directly on the first day after operation as low dose heparin as prophylaxis in the same procedure in all patients groups. Full therapeutic anticoagulation was routinely performed after the first control follow-up CT without recurrence, until then low dose heparin was given as prophylaxis.

Reviewer 2 Report

Dear authors, thank you for your article. It was very interesting to read your work.

In this paper, they conducted a retrospective study of a patients with cSDH. Potential risk factors for cSDH recurrence were analyzed. Some very interesting results were presented.

My suggestions/questions are:

In Material and methods - Radiological evaluation it was written "Two independent neuroradiologists analyzed the postoperative imaging"

Please write the initials of neuroradiologists in this sentence. In the case of two different measurements, how did you make a consensus? Please write it in the text.

In Table 1 - Type of hematoma is divided into hygroma, chronic and subacute. Considering the different causes and pathophysiology of the mentioned subgroups - define subgroups of hematoma types in the methods.

Is there a relationship between blood types and hematoma subtypes?

In Table 1 - In-hospital complication - was mortality related to surgery and complications. If not, mortality should be listed separately in Table 1, not in hospital complications.

Given that in Table 1 and also in Figure 1 you wrote the percentage/number of blood type and its relation to recurrence/no recurrence. Either show the data in Table 1 or as Figures. My suggestions are to show them as Figure 1 because the paper is based on this information.

How do you explain the main results of this research? What is your hypothesis?

Author Response

Dear authors, thank you for your article. It was very interesting to read your work.

In this paper, they conducted a retrospective study of a patients with cSDH. Potential risk factors for cSDH recurrence were analyzed. Some very interesting results were presented.

Author´s response: We thank the Reviewer for this comment.

My suggestions/questions are:

  • In Material and methods - Radiological evaluation it was written "Two independent neuroradiologists analyzed the postoperative imaging". Please write the initials of neuroradiologists in this sentence. In the case of two different measurements, how did you make a consensus? Please write it in the text.

Authors’ response:

We apologize for the confusion and thank the reviewer for this comment. The methods section had been expanded.

Two independent radiologists performed postoperative follow-up Imaging assessment; the first radiologist was usually a radiologist outside our institute where patients had their follow-up examinations. The second radiologist from our institute did the imaging during the outpatient presentation to us to decide on the further procedure. In case of disagreement whether it was recurrence or not, we took the decision of our neuroradiologist from our institute, because we have all the imaging (pre- und after surgery) and we have the comparison between all imaging.

In our manageable cohort, such incidents occurred in exactly three patients who had MRI as follow-up and no CT, so an intramodal comparison was conditionally assessable for the external colleague.

Changes to the manuscript:

Two independent radiologists performed postoperative follow-up Imaging assessment; the first radiologist was usually a radiologist outside our institute where patients had their follow-up examinations. The second radiologist from our institute did the imaging during the outpatient presentation to us to decide on the further procedure. In case of disagreement whether it was recurrence or not, we took the decision of our neuroradiologist from our institute, because we have all the imaging (pre- und after surgery) and we have the comparison between all imaging.

  • In Table 1 - Type of hematomais divided into hygroma, chronic and subacute. Considering the different causes and pathophysiology of the mentioned subgroups - define subgroups of hematoma types in the methods. Is there a relationship between blood types and hematoma subtypes?

Authors’ response:

We thank the reviewer for this comment. We have added the differences between the three forms to the method section. We found no correlation between hematoma subtypes and blood type. We have added this point in the results section.

Changes to the manuscript:

  • Methods section: We distinguished three subtypes of cSDH: Hygroma as hypodens on CT, intraoperative showing a liquor-typical color. The chronic heamatoma with the color as old oil intraoperative and iso- or hypodens on CT. And finally, the subacute form, here intraoperative fresh hematoma parts showed up as hyperdens on CT as well.
  • Result section: We found no relevant differences in subtypes of recurrent cSDH and blood type.

  • In Table 1 - In-hospital complication- was mortality related to surgery and complications. If not, mortality should be listed separately in Table 1, not in hospital complications.

Authors’ response:

We apologize for the confusion and thank the reviewer for this comment. The table section had been modified. Mortality was not surgically associated. Therefore, we have changed the table according to your suggestion. 

  • Given that in Table 1 and also in Figure 1 you wrote the percentage/number of blood type and its relation to recurrence/no recurrence. Either show the data in Table 1 or as Figures. My suggestions are to show them as Figure 1 because the paper is based on this information.

Authors’ response:

We apologize for the confusion and thank the reviewer for this comment. We have added new figure 1 and figure 2 with percentage and number of blood type und relation to cSDH/recurrence.

Changes to the manuscript:

We have added new figure 1 and figure 2 with percentage and number of blood type und relation to cSDH/recurrence.

Figure 1: Distribution of blood type in the recurrence group of cSDH

Figure 2: Distribution of blood type in the none-recurrence group of cSDH

  • How do you explain the main results of this research? What is your hypothesis?

Authors’ response:

The hypothesis was whether there is an association between recurrence of cSDH and blood group, our work could prove this association. We tried to find the reason by literature-research; we could not find any work that could explain this association. We assume that perhaps there are factors at the cellular level that favor this association, such as receptors in blood type O, but this is much more suggestive than proven. We think our work provides a basis for such studies at the molecular and cellular levels. In a rat model of cSDH, the balance between levels of pro-inflammatory cytokines (IL-6, IL-8 and TNF-α) in the early stages and levels of anti-inflammatory cytokines (IL-10, IL-13) in the subacute and chronic phases influenced the introduction and absorption of the hematoma; among these, the upregulation of vWF is associated with recovery from cSDH [16]. While blood type O is associated with lower plasma vWF levels [11], we do not know if, for example, it could be associated with a decrease of vWF levels in the subdural cavity and a higher risk of cSDH recurrence.

Changes to the manuscript:

Discussion section: The hypothesis was whether there is an association between recurrence of cSDH and blood group, our work could prove this association. We tried to find the reason by literature-research; we could not find any work that could explain this association. We assume that perhaps there are factors at the cellular level that favor this association, such as receptors in blood type O, but this is much more suggestive than proven. We think our work provides a basis for such studies at the molecular and cellular levels.

Reviewer 3 Report

Hamed et al report in manuscript D: brainsci-2303352 an association between type O blood and recurrence of cSDH in a single center in Germany.

The rationale for this interesting study is based on previous observations of reduced thrombotic risk, lower vWB levels and the interaction with platelets and specific functions of platelets with type O blood.

Type O blood is the most prevalent blood type and varies among geographical locations (about 38% in Germany).

The figure presented is somewhat confusing as it compares the % of recurrence and no recurrence of SDH for each blood type separately. The better way of presentation is to compare % of recurrence among the groups. For example, showing that type O has x percent, type A y%..... recurrence, and Chi Square for significance.

Author Response

Hamed et al report in manuscript D: brainsci-2303352 an association between type O blood and recurrence of cSDH in a single center in Germany.

The rationale for this interesting study is based on previous observations of reduced thrombotic risk, lower vWB levels and the interaction with platelets and specific functions of platelets with type O blood.

Type O blood is the most prevalent blood type and varies among geographical locations (about 38% in Germany).

                Author´s response: We thank the Reviewer for this comment.

  • The figure presented is somewhat confusing as it compares the % of recurrence and no recurrence of SDH for each blood type separately. The better way of presentation is to compare % of recurrence among the groups. For example, showing that type O has x percent, type A y%..... recurrence, and Chi Square for significance.

Authors’ response:

We apologize for the confusion and thank the reviewer for this comment. We have added new figure 1 and figure 2 with percentage and number of blood type und relation to cSDH/recurrence.

Changes to the manuscript:

We have added new figure 1 and figure 2 with percentage and number of blood type und relation to cSDH/recurrence.

Figure 1: Distribution of blood type in the recurrence group of cSDH

Figure 2: Distribution of blood type in the none-recurrence group of cSDH

Round 2

Reviewer 3 Report

I thank the authors for their replies to my comments.